# A Deep Learning Model Based on Capsule Networks for COVID Diagnostics through X-ray Images

**DOI:** 10.3390/diagnostics13172858

**Published:** 2023-09-04

**Authors:** Gabriela Rangel, Juan C. Cuevas-Tello, Mariano Rivera, Octavio Renteria

**Affiliations:** 1Facultad de Ingeniería, Universidad Autonoma de San Luis Potosi, San Luis Potosi 78290, Mexico; gabriela.rangel@tecsuperiorslp.edu.mx; 2Tecnologico Nacional de Mexico/ITSSLPC, San Luis Potosi 78421, Mexico; 3Centro de Investigacion en Matematicas, Guanajuato 36000, Mexico; mrivera@cimat.mx (M.R.); octavio.renteria@cimat.mx (O.R.)

**Keywords:** deep learning, capsule network, dilation rate, convolution, COVID-19

## Abstract

X-ray diagnostics are widely used to detect various diseases, such as bone fracture, pneumonia, or intracranial hemorrhage. This method is simple and accessible in most hospitals, but requires an expert who is sometimes unavailable. Today, some diagnoses are made with the help of deep learning algorithms based on Convolutional Neural Networks (CNN), but these algorithms show limitations. Recently, Capsule Networks (CapsNet) have been proposed to overcome these problems. In our work, CapsNet is used to detect whether a chest X-ray image has disease (COVID or pneumonia) or is healthy. An improved model called DRCaps is proposed, which combines the advantage of CapsNet and the dilation rate (dr) parameter to manage images with 226 × 226 resolution. We performed experiments with 16,669 chest images, in which our model achieved an accuracy of 90%. Furthermore, the model size is 11M with a reconstruction stage, which helps to avoid overfitting. Experiments show how the reconstruction stage works and how we can avoid the max-pooling operation for networks with a stride and dilation rate to downsampling the convolution layers. In this paper, DRCaps is superior to other comparable models in terms of accuracy, parameters, and image size handling. The main idea is to keep the model as simple as possible without using data augmentation or a complex preprocessing stage.

## 1. Introduction

The year 2019 was a turning point worldwide due to COVID-19 (coronavirus disease), which paralyzed the world. Because the Severe Acute Respiratory Syndrome (SARS-CoV-2) was a new virus, no vaccines were available. At that time, an early and reliable diagnosis was vital to (i) isolate the infected persons and prevent further infections and (ii) give the proper treatment to reduce the mortality.

At that time, Reverse Transcription-Polymerase Chain Reaction (RT-PCR) testing was the most widely used approach to correctly diagnose a COVID patient. The test consists of specimens usually collected from their noses or throats and then sent to laboratories to generate a diagnosis.

On the other hand, there are some examples where computed tomography (CT) is used in emergency radiology for a reliable diagnosis: strokes, bone fractures, abdominal emergencies (trauma, small bowel occlusion, intussusception), or chest emergencies [1]. Chest imaging is a critical part of emergency radiology; most studies have focused on two topics: pulmonary embolism and pneumonia.

For these reasons, CT and X-ray images were the other approaches used for the early diagnosis and treatment of COVID-19 disease [2,3,4,5]. X-ray images are cheaper and faster than CT. In addition, patients are exposed to lower radiation levels on X-rays than on CT images. However, the diagnosis based on this approach has some issues to be improved. For example, expert radiologists must interpret these medical images in detail, which slows the process [6]. Additionally, the images can overlap other lung infections (pneumonia, tuberculosis, or bronchitis), making the diagnosis of COVID-19 challenging. Another problem is the limited number of images available with the disease to train more people.

These problems have motivated scientists from different areas to contribute their knowledge to help in their branch of expertise. The Deep Learning (DL) field was not the exception. Convolutional neural networks (CNN) are the core of DL because their algorithms specialize in obtaining and understanding complex patterns in images. Once the CNN is trained, the inference is in real-time. These skills are essential in medical image classification. There are several publications that use CNN for image diagnostics; e.g., detection of lung cancer [7], brain tumors [8], and Alzheimer’s disease [9]. In particular, Refs. [10,11,12] reports a procedure for the detection of breast cancer based on computerized tomography scans and [13] reports a method for identifying genetic disorders by analyzing facial gestures. So researchers are making efforts to develop new CNN-based image classification algorithms to detect COVID based on chest X-rays.

Despite their extensive use, CNN have some limitations. For example, CNN require large labeled datasets during training [14,15]. A commonly used solution is the data augmentation technique, which consists of artificially increasing the number of images in the dataset by using different augmentation techniques, such as geometric transformations (e.g., flipping, rotation, scaling, cropping, translation, Gaussian noise addition) or advanced augmentation techniques based on deep learning [16]. However, the data augmentation process requires a great amount of computational effort; this can be translated into additional memory and computational constraints. Also, it is important to note that, for some problems, data augmentation should not be performed because it may significantly change the content of the information (e.g., voice data, disease images, among others), which is the case for COVID X-rays images. Additionally, CNN are prone to losing spatial information among features due to the max-pooling operation. Several researchers argue that the use of this operation presents some disadvantages, such as: (i) if the object to be detected is very small, after the max-pooling operation, the size of the pixel will be further reduced, making it more difficult to be detected [17], (ii) by reducing the number of parameters from one layer to another, important information about the spatial relationship of the components is lost, and the focus will only be on the presence or absence of features [18], (iii) losing the spatial relationship of objects requires more training images and will force the network to use tools such as data augmentation to improve its performance [19], (iv) the pooling operation can provide a little translation invariance, but it will lose the precise location information of the features [20]. In addition, researchers have discovered an intriguing phenomenon called an adversarial example. This phenomenon consists of CNN easily cheating with a slightly modified test image (known as adversarial example) [21], which forces the architecture to misclassify with high accuracy [22].

A novel approach that addresses all of these limitations is called Capsule Networks (CapsNets). These networks were introduced by Sabourn et al. in 2017 [23], to improve how the network passes information through their layers. CapsNets try to mimic the inverse graphics process in the brain; thus, it is necessary to encode a large amount of information. These networks analyze images more complexly by switching from scalar operations to vector operations. This complex analysis allows us to obtain good classification results with few training images. In addition to classifying correctly, the CapsNet approach includes a stage in its architecture responsible for reconstructing the input image. In this way, the algorithm verifies that the values obtained in its training learn the most crucial image patterns.

In addition, the CapsNets approach offers promising results in several areas, such as robustness to affine transformations, the ability to segment highly overlapping digits [23], the ability to generate new data already labeled [24,25], better resistance to adversarial attacks [26,27]. Furthermore, CapsNets achieve state-of-the-art accuracy in the MNIST data set without the need for data augmentation and use fewer parameters than CNN [25,28].

According to the literature, the promising results previously mentioned were obtained with simple image data sets. However, CapsNets are limited when analyzing complex images. The architecture struggles to understand the entire context of the image, generating a large number of parameters, which results in substantial computational effort [20]. For these reasons, researchers have focused on combining the advantages of CNN with those of CapsNets, as shown in Table 1. For example, Yang et al. propose the RS-CapsNet network [20], which uses some ideas from the ResNet architecture and the Squeeze and Excitation block, both of which were ILSVRC winners [20,29]. The experimental results show that RS-CapsNet performs better on the CIFAR10, CIFAR100, SVHN, FashionMNIST and AffNIST datasets. It can also provide better translation equivariance, with fewer trainable parameters (65.11%) compared to the baseline CapsNets architecture. Despite the optimistic preliminary results of the CapsNets approach, there is still work to be done to achieve state-of-the-art results on complex datasets.

Recently, some papers used CapsNets combined with CNN for medical diagnostics. Mobiny and Van Nguyen [7] used CT chest scans for diagnostic lung cancer, using images of 32 × 32 pixels, and considered two classes: nodule and non-nodule obtaining. Their network achieves an accuracy of 88.55% with only 226 images and struggles with the reconstruction stage; for that reason, they add a convolutional decoder. Afshar et al. [8] used CapsNets to diagnose the type of brain tumor; they trained with 3064 MRI images with a small resolution (64 × 64 pixels) and obtained a classification accuracy of 78% with three tumor classes: Meningioma, Pituitary, and Glioma. Kruthika et al. [9] proposed a CBIR system using 3D capsule network, 3D convolutional neural network, and pretrained 3D autoencoder technology for early detection of Alzheimer’s. They used MRI images with a size of 64 × 64 pixels for Alzheimer’s diagnosis with a classification accuracy of 94.06%. Xiang et al. [32] used a combination of CapsNet and ResNet for automated breast ultrasound tumor diagnosis. Their dataset contains 444 images of 128 × 128 pixels. They managed two classes (malignant or benign), achieving and obtaining an accuracy of 84.9%.

Respiratory diseases can also be detected by analyzing radiological images. Mittal et al. [33] used convolutions and dynamic capsule routing to diagnose pneumonia on 5857 chest radiographs with 100 × 100 resolution and obtained an accuracy of 95.9% to classify normal or pneumonia. Khanna et al. [4] developed the Detail Oriented Capsule Networks (DECAPS) model for the automatic diagnosis of COVID-19 using 746 chest CT images with a size of 448 × 448 pixels. In addition, they used GANs for data augmentation. Their model achieved 87.6% accuracy in detecting two classes: Patients with COVID-19 and non-COVID-19. Afshar et al. [34] achieved a detection accuracy of 95.7% in two classes using their COVID-CAPS model. They used training images of 224 × 224 pixels, and a transfer learning approach tuned with a new dataset constructed from an external dataset of X-ray images. Toraman et al. [35] proposed a convolutional CapsNet approach to detect COVID-19 disease from X-ray images using capsule networks. They used CT scan images with a size of 128 × 128 to detect three classes: COVID-19, no findings, and pneumonia. Also, they used the max-pooling operation and data augmentation. Their model achieved an accuracy of 84.22%.

In this paper, we present the design of a computational model called DRCaps for medical image diagnostics; i.e., detection of three classes: COVID-19, pneumonia, and healthy. We obtain the accuracy in the classification of 90%. The model is based on the CapsNet approach combined with a convolution stage and uses the stride and the dilation rate hyperparameters to avoid the max-pooling operation. Unlike the models mentioned above that use low-resolution images, our model handles images with larger resolution and uses the reconstruction stage as a regularization method. Furthermore, our model does not require data augmentation or any complex preprocessing tasks, keeping the architecture as simple as possible.

This paper is organized as follows. Section 2 explains the computational model architecture in detail, along with the dataset and all the adjustments involved in the model. Section 3 lists the results obtained by the models described in Section 2 and compares the performance of this model with other models based on CapsNets and CNN on image diagnostics. Finally, the paper shows a discussion of the results obtained, and the conclusions mention the general contributions of the study.

## 2. Materials and Methods

This section presents the dataset used, data preprocessing, describes the CapsNets original model and the proposed DRCaps model, the training and the hyperparameter selection procedures, and the experimental platforms.

### 2.1. COVIDx Dataset

We use the open-source COVIDx dataset [36], which is composed of different datasets such as: COVID-Chestxray dataset [37], COVID-19 Chest X-ray Dataset Initiative [38], COVID-19 Radiography Database [39], RSNA Pneumonia Detection Challenge [40], RSNA International COVID-19 Open Radiology Database (RICORD) [41], among others.

The version used for the experiments was COVIDx7A, which contains 16,690 images with three classes: pneumonia, healthy, and covid. Each image has a 1024 × 1024 size and, for simplicity, is represented in the RGB format with replicated channels. Figure 1 shows an example of each class in the COVIDx dataset.

The COVIDx dataset contains real images with high resolution, so the challenge in deep learning models is the training stage because it requires great memory capacity. For example, memory usage is minimal if we use the MNIST dataset [42]. Because MNIST has 60,000 training images and each image has a dimension of 28 × 28 pixels, it needs about 47 million data space. However, the COVIDx dataset, despite having only 15,000 training images, they have a resolution of 1024 × 1024, which requires a space of a little more than a million parameters per image, making it very difficult to load the entire dataset in memory. For these reasons, we decided to use an image size of 256 × 256 for images. Also, we decide to use the *Tf.data API* [43]. This API allows us to handle large amounts of data, read from different data formats, and perform complex transformations in a fast and scalable way. However, it requires us to organize the dataset so the API can handle it.

The original version of the COVIDx V7A dataset contained only two image folders called train and test; with two additional text files with the following information: patient id, filename, class, and data source for each image. However, the API used for the proposed model requires that each folder contain three subfolders with each class’s name and the corresponding images. Figure 2 shows the organization of the data set so that it can be used by the *Tf.data API*.

Finally, Table 2 shows the distribution of the data set in terms of classes (training and testing). For training, we have 15,111 images spread over 5474 cases of pneumonia, 7966 cases of healthy images, and 1670 COVID images. The training data set is split into 80% training (12,089 images) and 20% for validation (3022 images). In the case of testing, we have 1579 images: 594 with pneumonia, 885 healthy, and 100 with COVID.

### 2.2. Capsnet Baseline

The original CapsNets architecture is shown in Figure 3, and this architecture has only three layers: Conv layer, PrimaryCaps layer, and DigitCaps layer. Moreover, CapsNets have a Reconstruction stage formed by three FC layers. According to Sabourn [23], the input dataset used was the MNNIST dataset, which has ten classes and handles images of size 28 × 28 as we can see in Figure 3.

The Conv layer is used to extract the main features of the input image. The original Sabour architecture selects 256 channels, or filters, with a kernel of 9 × 9, with a stride of 1 and the ReLU function as shown in Figure 3.

So, given the 256 channels with the size of 20×20 from the conv layer, as shown in Figure 3, a kernel of 9×9 with a depth of 256 and a stride of 2 is applied, resulting in 32 PrimaryCaps layers each of size 6×6×8, where each PrimaryCap has eight dimensions (8D). These operations generate 1152 capsules. Each capsule has two components: magnitude and orientation. The magnitude represents the probability that the entity exists, while the orientation represents the instantiating parameters or properties of the entity.

Once the capsules are computed, the network decides which information will pass to the next layer. Due to the capsule approach, Sabour et al. propose new tools capable of handling data in vector form [23]. These tools are the nonlinear squashing function and the routing by agreement method.

As mentioned above, each entry of an output vector in a capsule represents the probability that the associated entity is present in the current input. So, it is necessary to use the nonlinear squashing function because only the length of the vector changes, not the orientation. Also, the squashing function obtains a vector with values 0 and 1; ensuring that small vectors take values close to 0 and large vectors get values below 1. Equation (Equation 1) shows the squashing function proposed by Sabour et al. [23], where $v_{j}$ is the vector output of the capsule *j* and $s_{j}$ is its total input.
(1)$$v_{j}=squash\left(s_{j}\right)=\frac{{∥s_{j}∥}^{2}}{1+{∥s_{j}∥}^{2}}\frac{s_{j}}{∥s_{j}∥}$$

According to Sabour et al. [23], for all but the first capsule layer, the total input to a capsule $s_{j}$ is a weighted sum over all prediction vectors ${\hat{u}}_{j|i}$ from the capsules in the layer below by multiplying the output $u_{i}$ of a capsule in the layer below by a weight matrix $W_{ij}$, which means ${\hat{u}}_{j|i}=W_{ij}u_{i}$ and $s_{j}=\sum_{i}c_{ij}{\hat{u}}_{j|i}$. This operation uses a weight transform matrix as shown in Figure 3, which encodes the spatial importance and other relations among the characteristics of the low-level capsules and the current one. If one of the calculated prediction vectors has a high value with a possible parent, then there is a downward feedback where the values of the coupling coefficients ($c_{ij}$) are adjusted to select the correct connection path by the iterative dynamic routing process, as explained in Algorithm 1. This results in a more intelligent selection than just choosing the most significant number, like in max-pooling.
**Algorithm 1:** Routing Algorithm, according to Sabour et al. [23]1:**procedure** Routing (${\hat{u}}_{j|i},r,l$)2:    for all capsule *i* in layer *l* and capsule *j* in layer (l+1): $b_{ij}\leftarrow 0$.3:    **for** *r* iterations **do**4:        for all capsule *i* in layer $l:c_{ij}\leftarrow softmax(b_{i}$) ▹softmax computes Equation (Equation 2)5:        for all capsule *j* in layer (l+1): $s_{j}$←$\sum_{i}c_{ij}{\hat{u}}_{j|i}$6:        for all capsule *i* in layer (l+1): $v_{j}$←$squash(s_{j})$▹squash computes Equation (Equation 1)7:        for all capsule *i* in layer *l* and capsule *j* in layer (l+1): $b_{ij}$←$b_{ij}$ + ${\hat{u}}_{j|i}$$v_{j}$8:        **return** $v_{j}$

The coupling coefficients between the capsule *i* and all the capsules in the layer above sum up to 1 and are determined by a routing softmax function, whose initial logits $b_{ij}$ are the log prior probabilities that the capsule *i* should be coupled to the capsule *j* as shown as follows:(2)$$c_{ij}=softmax\left(b_{i}\right)=\frac{exp(b_{ij})}{\sum_{k}exp\left(b_{ik}\right)}.$$

We detail the complete dynamic routing by agreement process in Algorithm 1, where the parameter *r* is the selected iteration number and the value *l* is the number of the current layer.

Finally, the DigitCaps layer ends with ten 16-dimensional vectors, as shown in Figure 3, one vector for each digit. This layer is the final prediction and can produce two outputs. The first output consists of ten vectors produced by the DigitCaps Layer, where each vector corresponds to each class in the network. Then, this output uses the norm $L_{2}$ to calculate the length of each vector. Finally, the vector values are the confidence to detect the associated class, i.e., the prediction. The second output is the reconstruction stage.

### 2.3. DRCaps Model

Figure 4 describes the architecture of the proposed DRCaps model. Note that it combines convolution layers with capsule layers with three stages: (1) Convolutional, (2) Capsule, and (3) Reconstruction.

The Convolutional Stage aims to extract features in complex images using the dilation rate hyperparameter. The dilation rate with the stride hyperameters allows the omission of a max-pooling operation and improves the spatial relationship problem. The second stage includes PrimaryCaps and ClassCaps. At this stage, the agreement algorithm implements a dynamic routing [23]. The last Reconstruction Stage evaluates how the model encodes the input features; i.e., reconstructs the input image. Also, this stage works as a regularization parameter in the training. In the end, the model combines the ClassCaps output and the Reconstruction Stage for the optimization weights. The following subsections describe in detail how each stage is formed.

#### 2.3.1. Convolution Stage

The Convolutional Stage (CS) converts the intensity of the pixels into activity detectors of local features and uses them as input for the next stage. The CS is formed by seven convolutional layers, as shown in Figure 5. The layers conv1, conv3, and conv5 use the dilation rate. Dilation modifies the convolutional kernel by introducing spaces between the elements.

Figure 6 illustrates how the dilation operates: a 3 × 3 kernel with a dilation rate of (2,2) will have the same field of view as a 5 × 5 kernel while only using nine parameters. Imagine taking a 5 × 5 kernel and deleting every second column and row, resulting in a broader field of view at the same computational cost. So, the same number of parameters of a 3 × 3 kernel can cover the same region of a 9 × 9 kernel or a 17 × 17 kernel with different dilation rates. Kernel dilation is particularly popular in real-time segmentation, where it is needed to cover a wide field of view and cannot afford multiple convolutions or larger kernels [44].

In the DRCaps model, the conv1 layer applies 128 filters to an input image of 256×256 pixels, using kernels of 3×3 with stride equal to 1 and with a dilation rate equal to (8,8). This configuration generates an output size of 240×240. As the original goal is to reduce the size of the feature maps, the next convolutional layer (conv2) uses a bigger stride value and only 64 filters. This reduces the features maps size to 119×119. So, next layers change the stride or dilation rate value for better feature extraction. Finally, CS ends with 64 filters with a size of 12×12. In addition, all the convolutional layers use ReLU activation function. Table 3 summarizes the hyperparameter selection on each convolutional layer.

#### 2.3.2. Capsule Stage

The Capsule Stage (CaS) consists of two layers: PrimaryCaps Layer and ClassCaps Layer. The PrimaryCaps Layer is the lowest level of multidimensional entities from an inverse graph perspective. This corresponds to a reverse rendering process. Given the 64 filters with the size of 12×12 from the last convolutional layer, a kernel of 9×9 with a depth of 64 and a stride equals 2, yielding 32 PrimaryCaps layers each of size 2×2×8, where each PrimaryCap has eight dimensions (8D), as shown in Figure 7. These operations generate 128 capsules. Each capsule is a group of neurons that code the probabilities of feature detection in an output vector. This output vector has two components: magnitude and orientation. The magnitude represents the probability that the entity exists, while the orientation represents the instantiating parameters such as pose (position, size, orientation), deformation, and texture, among others.

The ClassCaps layer follows the PrimaryCaps layer. Thus the network ends up with three vectors of 16 elements each, one vector for each dataset class. This matrix works to get two outputs, as shown in Figure 4. The first output consists of three vectors produced by the ClassCap layer, where each vector corresponds to each class in the network. Then, this output uses the norm $L_{2}$ to calculate the length of each vector. Finally, the values of the vector are the confidence of detecting the associated class, i.e., the prediction. The second output is the reconstruction stage, which we will explain in the next section.

#### 2.3.3. Reconstruction Stage

Finally, the Reconstruction Stage (RS) uses the output of the ClassCap Layer as input for recreating the original input image. Then the model minimizes the distance (loss) between the reconstructed and original images. Note that, whereas a traditional CNN only cares about whether or not the model predicts the correct classification, CapsNets use the reconstruction stage as a regularization method to improve the results. In our architecture, this stage is formed by four fully connected layers with 64, 128, 128, and 65,536, as shown in Figure 8. It is also important to note that, in the reconstruction stage, specifically, the decoder is part of the network that could generate more parameters due to their fully connected layers. In the DRCaps model, this stage generates a total parameter of 8,482,112. For these reasons, some models prefer to leave this stage out of their architectures, mainly if they handle RGB complex images.

Table 4 summarizes the hyperparameters used in the DRCaps model. The table explains all the hyperparameters necessary to replicate the model.

#### 2.3.4. Loss Functions

We train the DRCaps model by minimizing a total loss that includes two loss functions. According to Sabour, the first loss is the margin loss that represents the probability that a capsule entity exists based on the length of the instantiation vector [23]. The second loss (RS) promotes correct reconstructions of the input data. We use this function to encourage the capsules to encode the instantiation parameters of the input class; this loss acts as a regularizer. Following, we present the details of the particular losses.

##### Margin Loss

The margin loss function, Equation (Equation 3), was proposed by Sabour in the original CapsNet paper. According to Sabour [23], $T_{k}$ is equal to one if the diagnostic of class *k* is present and $m^{+}$ = 0.9 and $m^{-}$ = 0.1. We use ϵ = 0.5. The total margin loss is the sum of the losses of all class capsules.
(3)$$L_{k}=T_{k}max(0,m^{+}-\|v_{k}{\left\|\right)}^{2}+\epsilon \left(1-T_{k}\right)max(0,\|v_{k}{\left\|-m^{-}\right)}^{2}$$

##### Reconstruction Loss

The reconstruction loss corresponds to the mae, see Equation (Equation 4). This loss function takes the difference between the model prediction and the ground truth, applies the absolute value to that difference, and then averages it across the entire dataset. Therefore, all errors will be weighted on the same linear scale.
(4)$$mae=\frac{1}{n}\sum_{i=1}^{n}\left|y_{i}-x_{i}\right|$$

### 2.4. Training the DRCaps Model

Figure 9 describes the DRCaps model training process. First, the input image is passed through the DRCaps model and generates the ClassCap layer, as shown in Figure 4. The model then uses a total loss function formed by two different functions: margin loss and reconstruction loss. The first is the function obtained from the model’s prediction and the actual target. The model’s prediction corresponds to the vector with the highest magnitude in the ClassCap layer. The second loss compares the image reconstruction (decoder output) and the original input. The reconstruction loss is multiplied by a λ hyper-parameter that weights the relative contribution of each term. Hence, it is summed with the margin loss to obtain the total loss, Figure 9.

Our implementation uses images to a size of 256×256 and normalize, λ=40, a batch size equal to 32, and the Adam optimizer with the default parameters: a learning rate, *lr* = 0.001, and a *lr decay* = 0.9. Table 5 summarizes the hyperparameters

### 2.5. Experimental Platform

The experiments were carried out on a server at CIMAT called Tinieblas. The integrated development environment is JupyterHub. The Tinieblas server has three GPU cards: two GeForce RTX 2080 Ti with 11 GB VRAM and a GeForce RTX 2080 with 8 GB VRAM, which has a Compute capability (CC) equal to 7.5. The deep learning framework used was Keras 2.5.0 with TensorFlow 2.5.0 as a backend, as shown in Table 6.

## 3. Results

We started replicating the CapsNet baseline on the MNIST dataset. As shown in the first row in Table 7. Then, we decided to probe two different configurations (CapsNet V2 and V3) with the dilation rate parameter on the same MNIST dataset. With these experiments we can observe how the dilation rate reduces the capsule number and the total parameters. Whereas, the accuracy is increased.

Then, we decided to use the CaspsNet V3 on the COVIDx dataset. However, since we used a complex image with a depth of 3, the number of capsules and parameters increases significantly. Even when we decided to use only a depht of 1. These increases in the number of capsules and the number of parameters generate great computational complexity in the dynamc routing by agreement method. This results in computer memory saturation and makes it impossible to complete training, as shown in the four and five rows in Table 7.

So, it was necessary to reduce the number of capsules and parameters in order to train the model successfully. We reduce the FC layers in the reconstruction stage and remove the dilation rate. The new model (CapsNet V4) has three convolutional layers with a stride equal to 2 in each layer. Then we decided to change the λ hyperparameter, which had favorable results. In addition, we decided to use only one channel in the input images (CapsNet V5). This small change allows us to reduce the number of parameters and improve the accuracy, as shown in Figure 7. Also, we decided to try different configurations in the CS, such as kernel sizes, added the dilation rate hyperparameter, and changed the stride value. One thing that caught our attention was that as we increased the number of capsules, the training time per period increased, while the accuracy decreased. For example, we designed a configuration that generated 3200 capsules and used a training time of two and a half minutes per epoch, achieving an accuracy of 77%, while the other configuration with only 288 capsules achieved an accuracy of 81%.

Finally, we come up with the final DRCaps model; see Figure 4. Table 4 summarizes this model. This model generates only 128 capsules and handles 10,192,128 parameters, achieving an accuracy of 88%, as shown in the second column of Table 7. Finally, we tested four different versions, Table 7 summarizes the model results. The first version (V1) kept the parameters mentioned afterward. In the second version (V2), we used λ=40. In the third version (V3), we increase λ=80, and the last version uses λ=0.328. As shown in Table 7, the second version is the configuration that offers the best result.

Figure 10 shows the training loss functions of the DRCaps model up to 50 epochs. However, we can observe that only 20 are enough to obtain a good performance. The red line is the decoder loss (mae), the green line allows the loss of CapsNet (margin loss), and the blue line shows the total loss function, which is formed by adding the loss of CapsNet and the loss of the decoder increased by the reconstruction value of λ.

Figure 11 shows some examples of the output of the reconstruction stage in DRCaps V2. These results correspond to λ = 40, which produced an accuracy of 90%. The first four rows of Figure 11 are random examples of the images inputted into the model, and the last four rows show how the network attempts to reconstruct them. As can be seen, the network focuses on the chest section and attempts to recover the size, position, and shape of the lungs. On the other hand, Figure 12 shows the output of DRCaps V4. In this case, the regularization parameter λ is small and does not enforce a correct reconstruction of the input images.

## 4. Discussion

One of the main problems with CapsNets is that they struggle when handling complex images because the network wants to understand everything about the input image. Often, the region of interest in the input image to analyze to produce a correct classification is a small fraction. This results in very time-consuming training and a decrease in accuracy.

For these reasons, convolutional networks continue to be used as the first stage of the model for feature extraction, as they have demonstrated their efficient performance in image management. The selection of parameters in each convolutional layer is essential because the information obtained is the one that will form the capsules in the next stage. For example, if we do not reduce the size of the feature maps resulting from the CS, when entering the CaS, the number of capsules will be very large. Some articles that work with large images use the max-pooling operation to do this size reduction, even though it causes a loss of information. In our model, instead of using this operation, we decided to use dilation convolution (controlled by the dilation rate parameter) to cover a broad field of the image at a lower computational cost, in addition to using the stride parameter to reduce the size of the feature maps.

At the beginning of our experiments, we only used the dilation rate parameter, which generated large feature maps in the last convolutional layer; this caused the network to have many capsules, which caused slow training and low accuracy. The reason may be that the capsules tried to codify and reconstruct all the details of the image, but much of that information is irrelevant to our task. According to the experiments, there is a relationship between the number of capsules and the complexity of the images to analyze. For example, for the original CapsNet architecture, which uses a 28×28 image size, their model can handle 1152 capsules with reasonable accuracy. However, when we tried to use an excessive number of capsules (18,432; 2000; 67,712) the accuracy dropped to 60%, 71%, and 10%, respectively. For these reasons, we reduced the capsule number by manipulating the convolutional kernel stride to improve accuracy. Figure 5 and Table 3 show how the dilation rate and the stride reduce the size of the filters in the CS. This results in creating a smaller number of capsules and a smaller number of total parameters in the network.

In the CaS, we maintain the same arbitrary parameters as the original paper, such as the dimension of the capsules equal to eight, the size of the ClassCap layer as 16D capsule per class, and three dynamic routing iterations. We tried to adjust this parameter by hand with no significant results.

Finally, we observed that RS requires a larger number of parameters (weights) in the network, and some implementations (reported works) prefer to omit this stage. However, this stage is essential for regularizing the process; it enforced to feature extractor to codify all the information in the original image. Our observation is consistent with reported works that modified RS to improve accuracy [23,24,25,45,46]. We noted that the weight λ of the reconstruction loss significantly affects the result of our model and prevents overfitting. Furthermore, the experiments show a correlation between the input image size and the value λ: we set it equal to the proposed value of 0.0005 (original paper) multiplied by the size of the images used. Once you have this reference value, you can play with the values up or down, observing the behavior of the network. From Table 7, we can see that, in our case, the reference value with which we started was 32,768, we tried to increase such a value a little to 40 observing good results, so we decided to double the value to 80 and produced a decrease in the accuracy obtained. We trained our model with a modest number of examples that rely on the experimental evidence that CapsNets can generate new labeled data from existing data in a more realistic way than the data augmentation process [24].

Table 8 shows the advantages obtained with the DRCaps model and compares the model with similar architectures. The architectures in Table 8 are ordered with greater accuracy. Although our model did not reach the first place in Table 8, we can say that it offers a more robust architecture. For example, the first place in the table uses only two classes and uses noise-removal methods as image preprocessing. The second place in the table uses images smaller than ours and only has two classes, simplifying the task’s complexity. The third place at the table handles similar image sizes, but they use only two classes and do not include a reconstruction stage because they use color images. Also, they still use the max pooling operation on the CS. Despite the Kruthika et al. model achieving an accuracy of 94.06% using three classes, their input images have smaller sizes and can such a larger number of capsules than ours. That is a limitation for implementing a detector of other diseases where the features would present high spatial frequency characteristics or more classes need be detected.

The articles that follow our architecture in Table 8 mention that data augmentation results in an acceptable accuracy, and others still use the max-pooling operation. Also, some articles say that with their computing resources, it is impossible to use CaspsNets for high-quality images. For the same reason, other papers skip the reconstruction stage to design a lighter model. In addition, few publications use CapsNet with image sizes greater than 128 × 128 pixels.

## 5. Conclusions

Computed tomography and radiography play a vital role in early diagnosis and treatment. There are many AI applications in medical diagnostics with image processing. Most of these are based on CNN. However, CNN has limitations, and nowadays there are more options to improve these models. For example, the CapsNet architecture is more robust to affine transformation, works well for a small dataset, and retains the spatial relationship between features. Additionally, the model uses the reconstruction stage as a regularization method to avoid overfitting.

Therefore, we propose the DRCaps model to classify chest X-ray for different medical diagnoses: healthy, pneumonia, and COVID. We improved the original CapsNet approach and proposed the DRCaps model, which combines the advantages of kernel dilation and CaspNets. Also, the proposed model uses fewer parameters, less data pre-processing, avoids the max-pool operation, and manages complex images. DRCaps model is verified through experiments obtaining and improving accuracy. The proposed model is simple, with low complexity and better results. According to the results, the number of capsules and the correct weighting of the reconstruction loss are critical factors to avoid overfitting and obtain good results with complex images. In addition, there are still many opportunities to improve the model. AI has the potential to improve medical image classification, so the proposed approach should be considered for the medical diagnosis of other diseases.

## Figures and Tables

**Figure 1 diagnostics-13-02858-f001:**
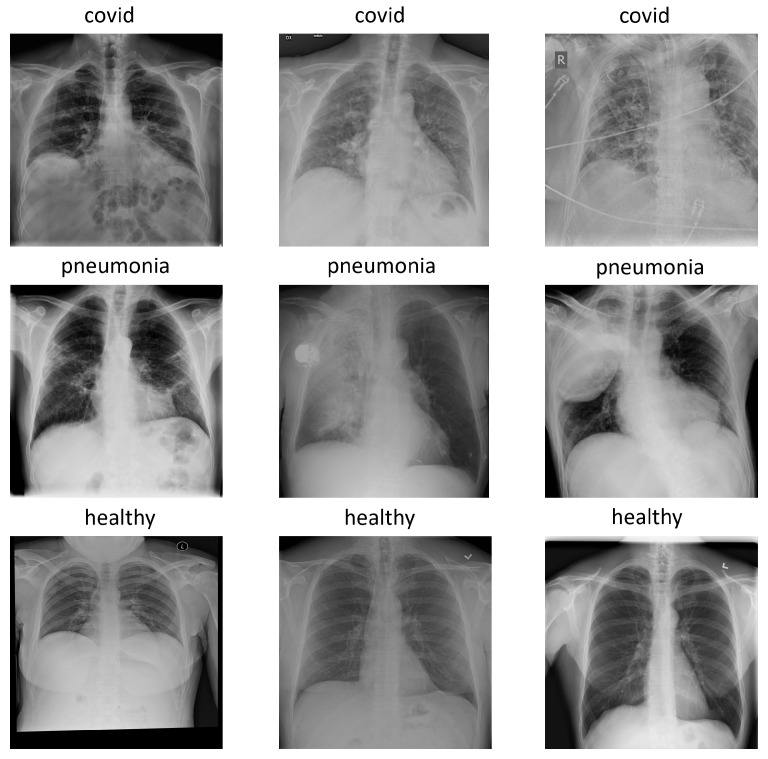
Examples of each class in COVIDx dataset: covid, pneumonia and healthy.

**Figure 2 diagnostics-13-02858-f002:**
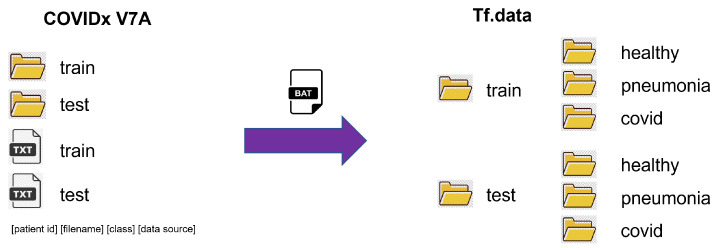
COVIDx dataset organization for *Tf.data API*.

**Figure 3 diagnostics-13-02858-f003:**
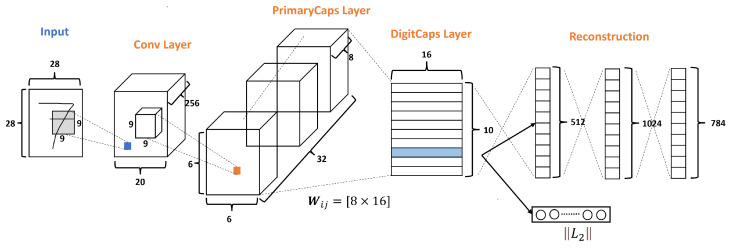
A simple CapsNet architecture with 3 layers, adapted from [23].

**Figure 4 diagnostics-13-02858-f004:**
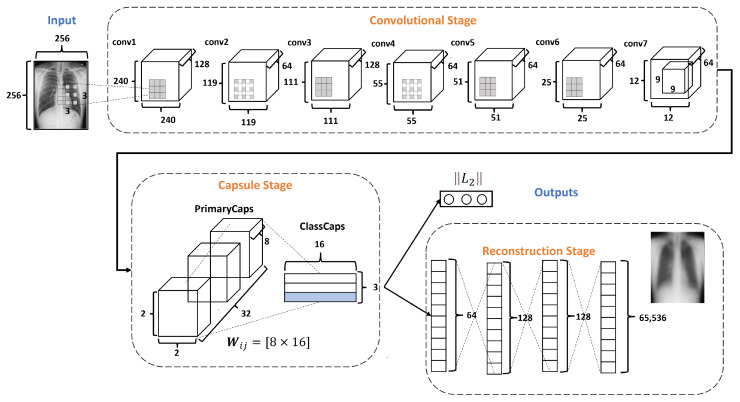
DRCaps model.

**Figure 5 diagnostics-13-02858-f005:**
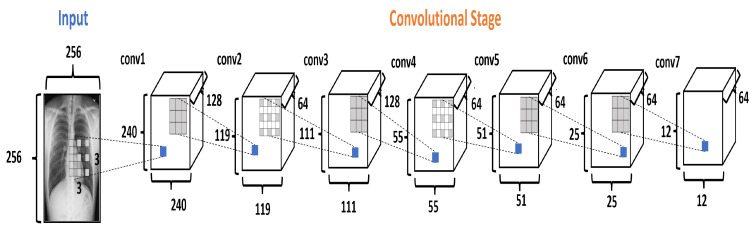
The Convolutional Stage in the DRCaps model.

**Figure 6 diagnostics-13-02858-f006:**
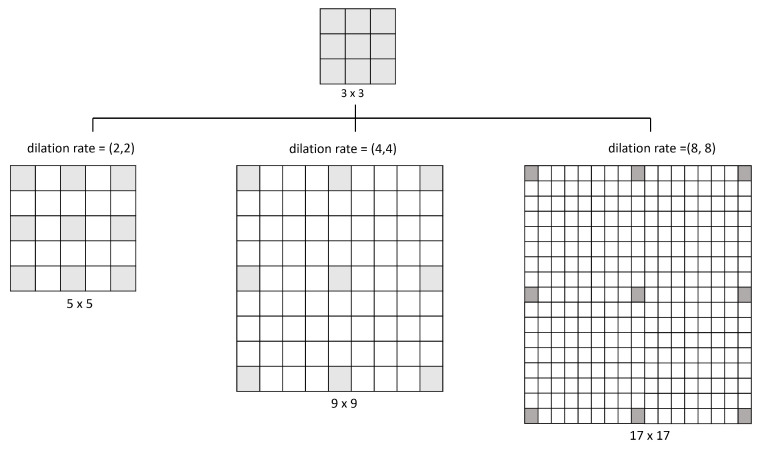
Examples of the space that covers different values of dilation rate used in this model.

**Figure 7 diagnostics-13-02858-f007:**
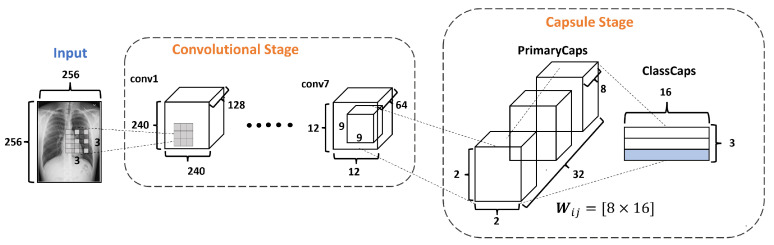
The Capsule Stage in the DRCaps model.

**Figure 8 diagnostics-13-02858-f008:**
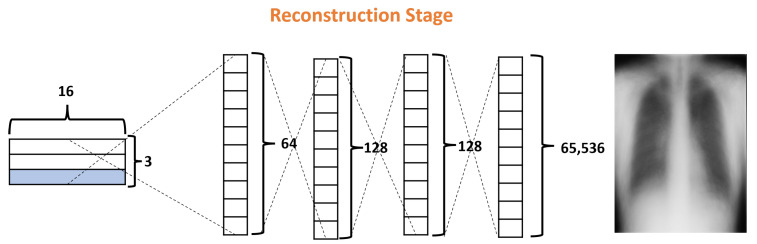
Decoder structure of the DRCaps model.

**Figure 9 diagnostics-13-02858-f009:**
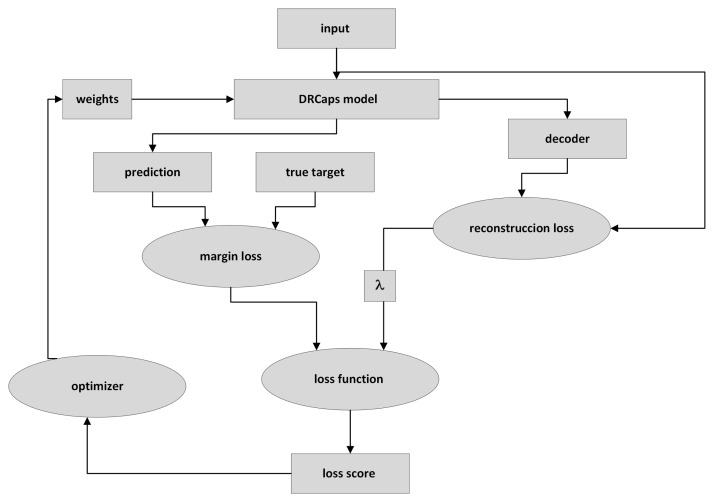
Training a CapsNet model.

**Figure 10 diagnostics-13-02858-f010:**
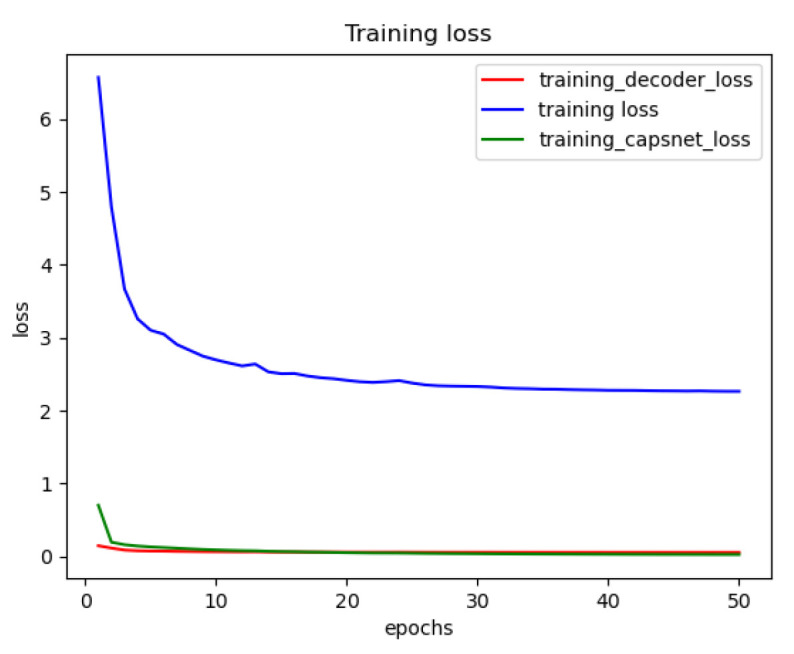
Reconstruction loss of the DRCapsModel.

**Figure 11 diagnostics-13-02858-f011:**
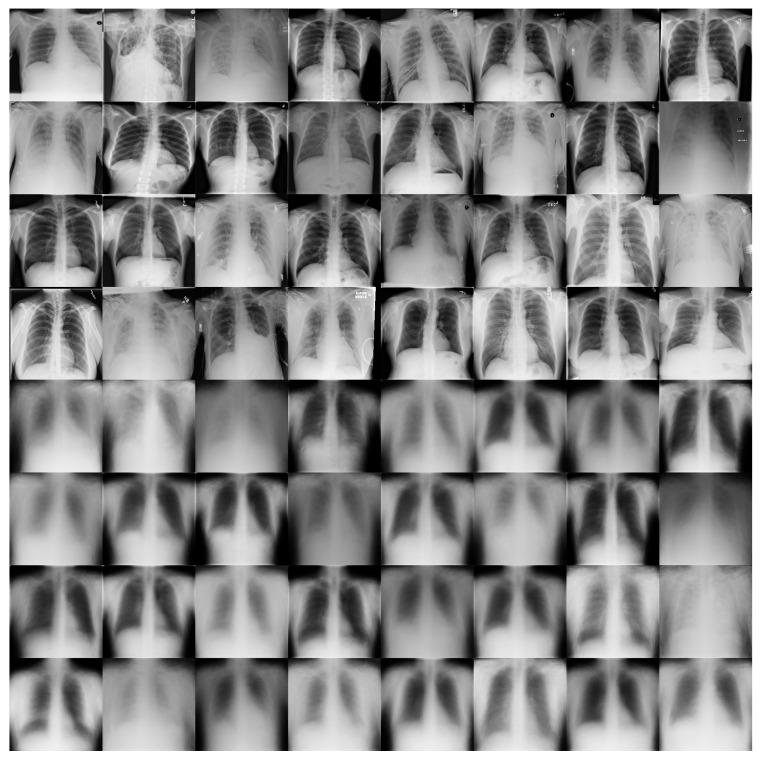
Reconstructed COVIDx images from RS in the DRCaps model with λ = 40. The first four rows are random examples of the images inputted into the model, and the last four rows show how the network attempts to reconstruct them.

**Figure 12 diagnostics-13-02858-f012:**
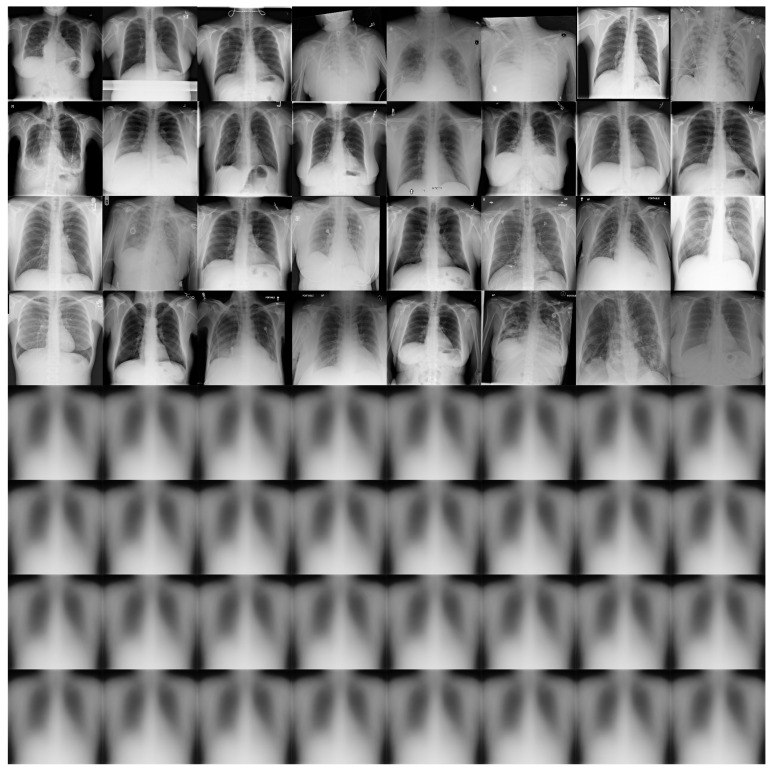
Reconstructed COVIDx images from RS in the DRCaps model with λ = 0.328. The first four rows are random examples of the images inputted into the model, and the last four rows show how the network attempts to reconstruct them.

**Table 1 diagnostics-13-02858-t001:** Examples combining CNN with CapsNets.

Network	Dataset	Combination	Ref.
DA-CapsNet	MNIST, CIFAR10, FashionMNIST, SVHN, smallNORB and COIL-20	Attention layers + CapsNet	[18]
FSSCaps-DetCountNet	The aerial elephant and The livestock	FSS classifier + CapsNet	[17]
DE-CapsNet	CIFAR-10, Fashion MNIST	SGE + CapsNet + DCNet++	[30]
RS-CapsNet	CIFAR10, CIFAR100, SVHN, FashionMNIST, and AffNIST	Res2Net + SE + CapsNet	[20]
ResCapsNet	LiDAR	ResNet + CapsNet	[31]

**Table 2 diagnostics-13-02858-t002:** COVIDx7A dataset classes.

Folder	Pneumonia	Healthy	COVID	Total
train	5474	7966	1670	15,111
test	594	885	100	1579

**Table 3 diagnostics-13-02858-t003:** Hyperparameter selection on each convolutional layers at the CS.

Layer	Filters	Kernel Size	Stride	Dilation Rate	Output Size	Parameters
conv1	128	3 × 3	1	(8,8)	240 × 240	1280
conv2	64	3 × 3	2	(1,1)	119 × 119	73,792
conv3	128	3 × 3	1	(4,4)	111 × 111	73,856
conv4	64	3 × 3	2	(1,1)	55 × 55	73,792
conv5	64	3 × 3	1	(2,2)	51 × 51	36,928
conv6	64	3 × 3	2	(1,1)	25 × 25	36,928
conv7	64	3 × 3	2	(1,1)	12 × 12	36,928

**Table 4 diagnostics-13-02858-t004:** Summary of hyperparameters of the DRCaps model.

Layer	Hyperparameters	
Input	Image size = (256, 256) channels = 1	
Convolutional	# Layers = 7 # Channels = 64,128 # filters = 7 filter size = 3,3 parameters = 333, 504	stride = 1,2 padding = valid dilation rate = (8,8),(4,4),(2,2) activation = ReLU
PrimaryCaps	# Capsules = 128 Capsule depth = 8 # Caps Layers = 32 parameters = 1, 327, 360	stride = 2 padding = valid filter size = 9
ClassCaps	# clases = 3 # Instantiation parameters = 16 parameters = 49, 152	
Reconstruction	# layers = 3 # layer sizes = 64,128,128,65536 activation = ReLU parameters = 8, 482, 112,	

**Table 5 diagnostics-13-02858-t005:** Training of the hyperparameters of the DRCaps model.

Arguments	Compile	Train
epochs = 50	optimizer = Adam	train images = 12,089 (80%)
learning rate = 0.001	learning rate = 0.001	validation images = 3022 (20%)
img width = 256	prediction loss = margin loss	batch size = 32
img height = 256	reconstruction loss = mae	steps per epoch = 378
*lr decay* = 0.9	metrics = accuracy	test images = 1579
routings = 3		
λ recon = 32.768		

**Table 6 diagnostics-13-02858-t006:** Experimental platform.

Server	OS	TensorFlow	Keras	GPU	CC
Tinieblas	Linux	2.5.0	2.5.0	RTX 2080 Ti	7.5

**Table 7 diagnostics-13-02858-t007:** Accuracy (acc) results at different hyperparameters configurations.

Network	Dataset	Image Size	Depth	Dilation Rate	Stride	Capsules	λ	RS	Parameters	Acc.
CapsNet baseline	MNIST	28 × 28	1	no	2	1152	0.392	5,121,024	8,215,568	98.5
CapsNet V2	MNIST	28 × 28	1	(2,2)	no	128	0.392	5,121,024	6,904,848	99.2
CapsNet V3	MNIST	28 × 28	1	(2,2), (4,4)	no	288	0.392	5,121,024	7,458,320	**99.6**
CapsNet V3	COVIDx	256 × 256	3	(2,2), (4,4)	no	438,048	0.392	5,121,024	375,963,520	-
CapsNet V3	COVIDx	256 × 256	1	(2,2), (4,4)	no	438,048	0.392	5,121,024	241,613,568	-
CapsNet V4	COVIDx	256 × 256	3	no	2	512	0.392	64,128,128	28,612,288	80.0
CapsNet V4	COVIDx	256 × 256	3	no	2	512	32.768	64,128,128	28,612,288	82.5
CapsNet V5	COVIDx	256 × 256	1	no	2	512	0.392	64,128,128	11,702,848	78.0
CapsNet V5	COVIDx	256 × 256	1	no	2	512	32.768	64,128,128	11,702,848	86.2
DRCaps V1	COVIDx	256 × 256	1	(8,8), (4,4), (2,2)	1,2	128	32.768	64,128,128	10,192,128	88.0
DRCaps V2	COVIDx	256 × 256	1	(8,8), (4,4), (2,2)	1,2	128	40	64,128,128	10,192,128	**90.0**
DRCaps V3	COVIDx	256 × 256	1	(8,8), (4,4), (2,2)	1,2	128	80	64,128,128	10,192,128	85.0
DRCaps V4	COVIDx	256 × 256	1	(8,8), (4,4), (2,2)	1,2	128	0.392	64,128,128	10,192,128	10.0

**Table 8 diagnostics-13-02858-t008:** Accuracy results of different CapsNets-based models in medical imaging. Some models do not have defined some parameter and are listed as not mentioned (nm).

Study	Model	Data	Input	Channels	Capsules	Classes	Decoder	Recons.	Max-Pooling	Data Aug.	Accuracy (%)
Ali et al. [47]	19-layers CNN	X-ray	227 × 227	3	nm	2	nm	no	yes	no	98.5
Mittal et al. [33]	ECC	X-ray	100 × 100	1	36,864	2	2	yes	no	no	95.9
Afshar et al. [34]	COVID-CAPS	X-ray	224 × 224	3	nm	2	nm	no	yes	no	95.7
Kruthika et al. [9]	CapsNets	MRI	64 × 64	1	18,432	3	3	yes	no	no	94.0
**this**	DR CapsNet	**CT**	**226 × 226**	**1**	**128**	**3**	**3**	**yes**	**no**	**no**	**90.0**
Mobiny and Van Nguyen [7]	Fast CapsNet	CT	32 × 32	1	2048	2	nm	yes	no	no	88.5
Khanna et al. [4]	DECAPS	CT	448 × 448	nm	nm	2	nm	no	no	yes	87.6
Sarki et al. [48]	VGG-16 CNN	X-ray	224 × 224	3	nm	3	nm	no	yes	no	87.5
Afshar et al. [8]	CapsNets	MRI	64 × 64	1	18,432	3	3	yes	no	no	86.5
Xiang et al. [32]	3-D ResCapsNet	ABUS	128 × 128	nm	nm	2	nm	no	no	nm	84.9
Toraman et al. [35]	CapsNets	CT	128 × 128	1	8192	3	4	yes	yes	yes	84.2

## Data Availability

Not applicable.

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
