# Peer review of "A Deep Learning Model Based on Capsule Networks for COVID Diagnostics through X-ray Images"

_diagnostics, 2023, doi:10.3390/diagnostics13172858_

Round 1
Reviewer 1 Report
In this paper, the authors propose a new model called DRCaps based on the CapsuleNet for X-ray image diagnosis, innovatively calculate the two types of losses ignored in previous studies, and obtain high diagnostic accuracy. I found some issues that need to be improved by the authors.
1. Please see Figure 1, please arrange the images according to their diagnostic classification.
2. Please see Figure 5, Table 5 and 3, Please use a consistent representation to represent the dilation rate.
3. Please see Table 5, digitCaps was never mentioned before, please check if it’s equal to classCaps .
4. In the model accuracy validation process, you can collect a set of data as the validation set to better illustrate the accuracy of the model.
5. Please see Table 9. Please list the model names used in each paper. And please keep the accurate results to at least one decimal place.
6. Please see Figure 9. Please extend the horizontal axis to prove that epochs equal to 20 is sufficient to minimize training loss.
Reviewer 2 Report
Dear Authors,
I have now concluded my review of the manuscript entitled "A Deep Learning Model Based on Capsule Networks for COVID Diagnostics through X-Ray images." The structure of the article is sound, effectively articulating the motivation behind the work, the key approach utilized (specifically the use of Capsule Networks and the proposed DRCaps model), and the results obtained from your experimental evaluation.
Despite the strong foundation of the paper, I believe that several areas could be further refined prior to publication.
1. The limitations of Convolutional Neural Networks (CNNs) are alluded to in the text, but a detailed exposition on these limitations is missing. Providing such information would offer readers a more precise understanding of the motivation behind the proposed solution.
2. The manuscript would be enhanced by a more thorough explanation of both CapsNet and the proposed DRCaps model. For instance, the rationale for combining the advantages of CapsNet with the dilation rate parameter in the DRCaps model was unclear.
3. While it is claimed that DRCaps surpasses the performance of other models, the paper would benefit from more specific comparisons. To illustrate, a paper titled "A CNN-Based Chest Infection Diagnostic Model: A Multistage Multiclass Isolated and Developed Transfer Learning Framework"(https://doi.org/10.1155/2023/6850772) claims to achieve superior accuracy using CNNs. A comparative analysis between your work and such previous studies would strengthen your argument.
4. You mentioned the incorporation of a reconstruction stage that ostensibly aids in preventing overfitting, as well as the decision to avoid the max-pooling operation for networks with convolution layers. However, the benefits of excluding max-pooling operations are not universally agreed upon in the current literature. Could you expound on the advantages of this approach in the context of your model?
I look forward to the revised manuscript.
Reviewer 3 Report
This paper develops a method DRCaps which combines CapsNet and CNN to detect COVID using X-ray images. I think the paper is interesting and relatively well written. The introduction explains the research gap clearly. The explanation of the method and the intuition of the DRCaps architecture is clear and reasonable. There are some innovations in the method. The figures are tables are informative. I have some comments for consideration.
Major comments:
1. The authors conducted several experiments to show the performance of the proposed method under different situations. I think the experiments lack comparison with baseline methods. So it is hard to be convinced that this method is better than existing methods. The paper can be improved by adding more experiments to demonstrate the advantages of the method with existing methods.
2. The authors mention that PCR testing for COVID is time-consuming, complicated, expensive, and limited by the supply of test kits. Using X-ray images do not have these disadvantages. My impression is that COVID testing becomes very cheap and convenient nowadays. And the testing doesn’t require a large database and computational resources. I am not fully convinced that using X-ray would be better than existing testing. I think some clarification considering the advantages of the proposed method is necessary. Will the proposed method provide result faster than existing test? Will the proposed method contribute to more general problems other than COVID-19 detection? I think elaborating on the significance and impacts of the proposed method would be helpful.
Minor comments:
1. This paper only uses accuracy to quantify the performance of the method. I suggest also including other common metrics.
2. Page 12 Line 307 referred 90% as the precision, other places talk about accuracy. Is this a typo? These need to be consistent.
3. The last few sentences of the abstract are not formatted.
4. I think the last three rows of Table 8 shows the accuracy (acc). I suggest adding some caption in Table 8 to let the readers know what acc is.
The writing is easy to follow.
Round 2
Reviewer 1 Report
No comments.
Reviewer 3 Report
Thanks for your efforts to revise the paper.